# Exploring the Advantages and Disadvantages of a Whole Foods Approach for Elevating Dietary Nitrate Intake: Have Researchers Concentrated Too Much on Beetroot Juice?

Alex Griffiths [1], Shatha Alhulaefi [2,3], Eleanor J. Hayes [2], Jamie Matu [1], Kirsten Brandt [2], Anthony Watson [2], Mario Siervo [4] and Oliver M. Shannon [2,*]

1. School of Health, Leeds Beckett University, Leeds LS1 3HE, UK; a.griffiths@leedsbeckett.ac.uk (A.G.); j.matu@leedsbeckett.ac.uk (J.M.)
2. Human Nutrition & Exercise Research Centre, Centre for Healthier Lives, Population Health Sciences Institute, Newcastle University, Newcastle upon Tyne NE2 4HH, UK; s.alhulaefi2@newcastle.ac.uk (S.A.); e.j.hayes2@newcastle.ac.uk (E.J.H.); kirsten.brandt@newcastle.ac.uk (K.B.); anthony.watson@newcastle.ac.uk (A.W.)
3. Department of Nutrition, Taif University, Taif 21944, Saudi Arabia
4. School of Population Health, Curtin University, Perth, WA 6845, Australia; mario.siervo@curtin.edu.au
* Correspondence: oliver.shannon@newcastle.ac.uk; Tel.: +44-1912081140

**Abstract:** In recent years, a number of studies have explored the potential salutary effects of dietary nitrate, with promising findings emerging. Indeed, numerous investigations have now demonstrated that increasing intake of dietary nitrate can reduce blood pressure, improve endothelial function, decrease platelet aggregation, increase cognitive function and brain perfusion, and enhance exercise performance. Most researchers have explored the health and/or performance effects of dietary nitrate by providing participants with concentrated beetroot juice, which is rich in this compound. Another strategy for increasing/optimising dietary nitrate intake, which could be embraced alongside or instead of nitrate-rich supplements in research and non-research settings, is the consumption of whole nitrate-rich vegetables. In this review, we explore the potential advantages and disadvantages of increasing consumption of various whole nitrate-rich vegetables to augment dietary nitrate intake. We compare the cost, convenience, availability, feasibility/acceptability, and efficacy of consumption of nitrate via whole nitrate-rich vegetables against concentrated beetroot juice 'shots' as defined supplements. We also discuss possible strategies that could be used to help individuals maximise their intake of nitrate via whole vegetables, and outline potential avenues for future research.

**Keywords:** nitrate; beetroot juice; health; exercise performance; supplementation; whole foods

## 1. Introduction

Dietary nitrate is an inorganic anion which, in the human body, serves as a substrate for the multifunctional signalling molecule nitric oxide (NO) [1]. Dietary nitrate is available from a variety of food sources, but most nitrate in the diet comes from vegetables, with smaller contributions coming from certain herbs, fruits, processed meat, and drinking water [2–4]. Although dietary nitrate intake can vary considerably between individuals, the average intake tends to be just over 100 mg/d in both healthy (median intake: 108 mg/d) and patient (median intake: 110 mg/d) populations [2]. Early data from observational and animal studies suggested that a high intake of dietary nitrate could be detrimental to health, increasing the risk of methaemoglobinaemia and certain types of cancer [5–7]. Consequently, an acceptable daily intake (ADI) of nitrate was established at 3.7 mg/kg/d [8]. However, other studies have questioned this view [9–11]. Indeed, recent studies have suggested that consumption of dietary nitrate alone does not cause methaemoglobinaemia [4], which, instead, is typically caused following ingestion or inhalation of an oxidising agent [12]. Likewise, some [13], although not all [14,15], recent

investigations have refuted the notion that nitrate intake increases cancer risk. In contrast to this early body of evidence, which tended to focus on nitrate as a potentially harmful dietary compound, research emerging in recent years suggests that increasing dietary nitrate intake (and thus enhancing NO bioavailability) can improve markers of cardiovascular and cognitive function. For example, dietary nitrate supplementation has been shown to reduce blood pressure (BP) by between ~2 and 10 mmHg [16–20], and could represent a potential adjunct to traditional antihypertensive medications. In addition, dietary nitrate supplementation has been shown to improve endothelial function [17,21], decrease platelet aggregation [17,19], and, albeit less reliably, enhance cerebral blood flow and cognitive function [22–24]. Over 100 studies have also been conducted exploring the potential benefits of dietary nitrate for sports performance, with findings from both a recent meta-analysis [25] and Delphi expert consensus [26] suggesting that nitrate can improve performance across a range of exercise modalities and in various populations. Several mechanisms have been proposed to explain these effects, including improvements in the efficiency of muscle contraction [27,28] and mitochondrial respiration [29], and modulation of tissue blood flow [30,31]. Interestingly, the health and performance benefits of nitrate are not apparent in individuals who use antibacterial mouthwash, which destroys bacteria in the mouth that play a crucial role in the processing of nitrate in the body [32,33].

Most studies investigating the health and/or exercise performance effects of dietary nitrate have administered this compound via nitrate-rich supplements, especially concentrated beetroot juice 'shots', which are highly enriched in nitrate and for which a nitrate-depleted placebo product is available [17,25,34,35]. For mechanistic research studies, the advantages to the provision of nitrate via concentrated beetroot juice are obvious, which may explain the ubiquity and dominance of this nitrate source in the literature. However, the translation of research that predominantly relies on the use of beetroot juice into the 'real world' could be limited if individuals are not willing or able to incorporate this particular product into their habitual diet. Another strategy for increasing/optimising dietary nitrate intake to influence human health and/or exercise performance, which could be used alongside or instead of nitrate-rich supplements in both research and 'real world' settings, is the consumption of whole nitrate-rich vegetables [36–38]. In this review, we explore the potential advantages and disadvantages of increasing consumption of various whole nitrate-rich vegetables to augment dietary nitrate intake, by focusing on the following questions:

1. Is it more cost-effective to consume dietary nitrate in the form of nitrate-rich vegetables or concentrated beetroot juice 'shots'?
2. Is it more convenient to consume dietary nitrate in the form of nitrate-rich vegetables or concentrated beetroot juice 'shots'?
3. Which is the more readily available source of dietary nitrate—nitrate-rich vegetables or concentrated beetroot juice 'shots'?
4. Is it more feasible/acceptable to consume dietary nitrate in the form of nitrate-rich vegetables or concentrated beetroot juice 'shots'?
5. What is the efficacy (in terms of eliciting health and/or performance effects) of dietary nitrate in the form of nitrate-rich vegetables versus concentrated beetroot juice 'shots'?

We also discuss some potential barriers to consumption of nitrate-rich vegetables, as well as opportunities for helping individuals optimise their intake of nitrate-rich vegetables, which may be of value in research and 'real world' settings.

## 2. Cost

Individual dietary choices are influenced by a number of factors, one of which is food pricing/cost [39]. Commercially available concentrated beetroot juice 'shots' are typically priced around GBP 2 or ~USD 2.50 each and contain a standardised nitrate dose of at least 400 mg for the most used product (Beet It Sport Shot, James White Drinks LTD, Ashbocking, UK). This may be prohibitively expensive for some individuals, if purchased/consumed daily. For example, in the United Kingdom (UK), the average household spend on food

and non-alcoholic drink in the financial year ending in March 2021 was GBP 69.20 per week [40] (equivalent to ~USD 86/week). The purchase of one beetroot juice 'shot' per day would, therefore, constitute roughly 20% of the average household spend on food and non-alcoholic drinks every week, and a much higher percentage in lower-income households. Two examples of how the cost might impact the intake of nitrate-containing foods are provided below.

### 2.1. Example 1: Vegetables Purchased from General Retailers in the UK

In developed countries and some developing countries, most consumers obtain vegetables from supermarkets supplied from large retailers with long-distance supply chains providing year-round supply. Table 1 shows the average dose of nitrate in some commonly consumed nitrate-rich vegetables, alongside the estimated cost to achieve a nitrate dose equivalent to the beetroot juice 'shot' specified above. As can be seen, it is possible to obtain 400 mg of nitrate at a relatively low cost via different whole vegetables. For example, for whole beetroot, the cost per 400 mg nitrate, based around the average nitrate content in this vegetable reported by Blekkenhorst et al. [41], is around $\frac{1}{4}$ the cost of a beetroot juice 'shot'.

**Table 1.** Quantity and cost of consuming a standardised nitrate dose (400 mg, equivalent to one beetroot juice 'shot') via nitrate-containing vegetables widely available in the UK.

| Food | Average Nitrate Content per 100 g of Each Vegetable [a] | Quantity Required to Consume 400 mg Nitrate [b] | Cost per 100 g of Each Vegetable [c] | Cost per 400 mg Nitrate [c] |
|---|---|---|---|---|
| Spinach | 193 mg | 208 g | GBP 0.43 (USD 0.53) | GBP 0.89 (USD 1.11) |
| Beetroot | 158 mg | 253 g | GBP 0.20 (USD 0.25) | GBP 0.51 (USD 0.63) |
| Iceberg lettuce | 94 mg | 424 g | GBP 0.22 (USD 0.27) | GBP 0.93 (USD 1.16) |
| Rhubarb | 110 mg | 364 g | GBP 0.75 (USD 0.93) | GBP 2.73 (USD 3.39) |
| Watercress | 232 mg | 172 g | GBP 1.45 (USD 1.80) | GBP 2.49 (USD 3.09) |
| Radish | 166 mg | 240 g | GBP 0.23 (USD 0.29) | GBP 0.55 (USD 0.68) |
| Arugula (Rocket) | 348 mg | 115 g | GBP 1.64 (USD 2.04) | GBP 1.89 (USD 2.35) |
| Celery | 187 mg | 214 g | GBP 0.32 (USD 0.40) | GBP 0.69 (USD 0.86) |
| Pak choi | 249 mg | 161 g | GBP 0.56 (USD 0.70) | GBP 0.90 (USD 1.12) |
| Kale | 175 mg | 229 g | GBP 0.42 (USD 0.52) | GBP 0.96 (USD 1.19) |
| Romaine Lettuce | 129 mg | 310 g | GBP 0.38 (USD 0.47) | GBP 1.18 (USD 1.46) |
| Swiss chard | 208 mg | 192 g | GBP 0.84 (USD 1.04) | GBP 1.61 (USD 2.00) |
| Turnip | 171 mg | 234 g | GBP 0.21 (USD 0.26) | GBP 0.49 (USD 0.61) |
| Fennel | 131 mg | 305 g | GBP 0.50 (USD 0.62) | GBP 1.52 (USD 1.89) |

[a] Nitrate content per 100 g of each vegetable adapted from [41], [b] Aggregate cost represents cost in 3 UK supermarkets (Sainsburys, Tesco, and Asda) per 100 g as of May 2023. Data rounded to three significant figures. [c] US dollar equivalent presented in parentheses and based on the available exchange rate in May 2023.

One major challenge is that the nitrate content of vegetables can be highly variable, depending upon factors such as plant genetics [42] and the way in which they were grown (e.g., soil conditions, sunlight, fertilisation) and stored (e.g., storage duration and temperature) [4,43]. Indeed, in the case of spinach—a commonly consumed nitrate-rich vegetable—Hord et al. [4] report a measured nitrate content in raw leaves varying between 23.9 and 387.2 mg/100 g. Thus, whilst the estimated cost of consuming 400 mg nitrate via spinach based around the average nitrate dose of 193 mg/100 g might be GBP 0.89 (~USD 1.11), the actual cost for this nitrate dose could be as little as GBP 0.44 (~USD 0.55) and as much as GBP 7.20 (~USD 8.94). After purchase, it also matters how the vegetables are prepared (e.g., washed, peeled, pickled) and cooked (e.g., boiled, steamed, fried).

### 2.2. Example 2: Vegetables Grown in Own Garden or Purchased from Local Markets

In developing countries, particularly in rural areas, the supply of perishable foods, such as leafy vegetables, tends to be more local; vegetables are grown in the consumer's

family garden or peri-urban vegetable farms, and harvested on the day they will be used, or dried in the sun for use in soups and stews [44]. While drying drastically reduces vitamin C content, it probably does not affect the nitrate content (per leaf). The cost is difficult to quantify, but is likely to be much lower than purchasing an imported product, such as beetroot juice.

In Table 2, we provide an outline of the nitrate content of some common vegetables grown or collected and available in markets in tropical and subtropical regions. It should be noted that many other vegetables sources of nitrate are likely to exist in these settings. However, for some of the most widely consumed leafy vegetables, such as sweet potato leaves, cassava leaves, Indian pennywort, and baobab leaves, few, if any, values on nitrate content have been published. Therefore, as discussed later, more research is needed to understand key vegetable sources of nitrate in different settings.

**Table 2.** Examples of nitrate-containing vegetables grown or collected and available in markets in tropical and subtropical regions, alongside the quantity required to consume a standardised nitrate dose (400 mg, equivalent to one beetroot juice 'shot').

| Food | Average Nitrate Content per 100 g | Quantity Required to Consume 400 mg Nitrate |
|---|---|---|
| Mustard greens [a] | 260 mg | 153 g |
| Swiss chard [a] | 208 mg | 192 g |
| Water spinach [a] | 145 mg | 276 g |
| Fenugreek leaves [a] | 137 mg | 292 g |
| Amaranth [a] | 112 mg | 357 g |
| Malabar spinach [a] | 102 mg | 392 g |
| Jute mallow [b] | 311 mg | 128 g |
| Fluted pumpkin [b] | 280 mg | 143 g |
| Bitter leaf [b] | 135 mg | 296 g |
| Roselle [b] | 128 mg | 313 g |

[a] From [41], and [b] from [45].

## 3. Convenience

A key strength of the commercially available beetroot 'shots' is their convenience. They are compact and easy to transport, which may be beneficial for individuals looking to 'top up' their nitrate intake away from home (e.g., when travelling) [46]. They also contain a specified nitrate dose, which reduces the burden on individuals. In contrast, the variable nitrate content of vegetables could make achieving a specific dose challenging (and, potentially, costly) on a case-by-case/day-by-day basis [4], as individuals will not know exactly how much nitrate is being consumed for each food/meal they eat. However, this may be less of an issue when looking at average nitrate intake over a period of weeks or months, if the nitrate content of vegetables is normally distributed, such that the variability may 'average out' over time.

From a research perspective, the standardised nature of the beetroot juice 'shots' allows researchers to ensure all participants receive the same nitrate dose [47]. This may be challenging with a whole-vegetable approach, given the variability in vegetable nitrate content could result in slightly different doses administered to each participant [4]. Similarly, the availability of a nitrate-depleted beetroot juice, which tastes, looks, and smells the same as the nitrate-rich equivalent, but is devoid of nitrate—something achieved by passing the juice through an ion-exchange resin that selectively extracts the nitrate—is particularly valuable as a placebo to researchers [48]. Not only does this allow researchers to isolate the impact of the nitrate on the research outcome of interest, but it also allows proper blinding of participants who may be able to discern their experimental condition with less refined placebos, such as fruit squash drinks.

Improved knowledge of the dietary sources of inorganic nitrate could make it easier for individuals to optimise their intake of dietary nitrate. For example, awareness of the types of vegetable that are typically high in nitrate may allow individuals looking to boost

their nitrate intake to prioritise the consumptions of such foods by swapping lower-nitrate vegetables with ones they know to be rich in this compound. The effectiveness of this strategy may be augmented by a recent development in the field by Qadir and colleagues [49], who demonstrated that by tightly controlling and manipulating the growing conditions of different vegetables, it is possible to produce particularly high- or low-nitrate vegetables. Specifically, differential fertilisation allowed the researchers to grow lettuce with higher (1060 mg nitrate/100 g) and lower (6 mg nitrate/100 g) nitrate concentrations. The growth of high-nitrate vegetables with a standardised minimum nitrate dose (if approved for commercial sale, e.g., as a high-nitrate salad bag) could be useful for individuals looking to boost their habitual nitrate intake. Similarly, this could be valuable for researchers as an alternative to the beetroot juice (active and placebo) 'shots'.

One final consideration from a convenience standpoint relates to the consumption of dietary nitrate prior to or during exercise to elicit ergogenic effects. In the days leading up to a competitive sporting event, research indicates that it is possible to elevate habitual nitrate intake and achieve ergogenic effects using both concentrated beetroot juice and whole vegetables [50–53]. Nevertheless, consumption of a nitrate-rich salad in the hours prior to exercise (e.g., to provide a final 'top up' dose of nitrate pre-competition), and during exercise, may be logistically difficult, due to the challenges of transporting enough of the vegetables to, or around, a sporting event. Similarly, the amount that must be consumed may be undesirably high for some individuals due to the perceived risk of stomach upset. On the other hand, beetroot juice 'shots', alongside other nitrate-containing supplements, such as nitrate-rich gels [54], may be particularly useful for consumption immediately pre- and/or during exercise, where they can be easily transported and consumed in relatively small amounts to achieve a potentially beneficial effect [46]. For example, a recent study from Tan and colleagues [46] showed that consumption of concentrated beetroot juice both before and during exercise helped to preserve elevated plasma nitrite concentrations, spare muscle glycogen, and limit the gradual rise in oxygen uptake during two hours of cycling compared with pre-exercise beetroot juice ingestion alone or placebo ingestion.

## 4. Availability

A range of nitrate-rich vegetables, such as those outlined in Table 1, are widely available in supermarkets and grocery stores in high-income countries, such as the UK and United States (US). Meanwhile, beetroot juice 'shots' can be purchased in many health food shops, in some supermarkets, and via the internet in both the UK and US. For individuals in high-income countries actively trying to increase their nitrate intake, availability is unlikely to be a major barrier to consumption for either nitrate vehicle. However, this may not be the case in other settings, such as low- and middle-income countries, where the intake of vegetables is often much less than 100 g per day [44].

High rates of hypertension are observed in Africa, where almost 50% of adults aged 25 years and above have elevated blood pressure [55]. Interestingly, Siervo et al. [56,57] demonstrated considerable potential for dietary nitrate to help combat hypertension in Tanzania, a country in Sub-Saharan Africa experiencing a hypertension epidemic. Specifically, 60 days of dietary nitrate supplementation reduced 24 h ambulatory systolic BP by around 10 mmHg, suggesting considerable potential for dietary nitrate as an anti-hypertension strategy in this setting. Although this study administered nitrate in the form of beetroot juice 'shots', which were provided to participants via the research team, this strategy may not be viable for many individuals in Tanzania or other lower- or middle-income countries outside of a research setting, given limited availability (alongside cost restrictions). On the other hand, the growth and purchase of native vegetables rich in nitrate (Table 2) could represent a vital tool to help combat the high rates of hypertension in such settings.

## 5. Feasibility and Acceptability

In contrast to the wealth of research exploring the efficacy of dietary nitrate (discussed further below), only a handful of studies have reported data on the feasibility and/or

acceptability of different strategies for augmenting dietary nitrate intake [34]. In one study by Ormesher et al. [58], who provided pregnant women with a single 70 mL concentrated beetroot juice 'shot' every day for 8 days, almost all of the participants (97%) indicated that they would consume beetroot juice 'shots' again, if they were experiencing benefits. However, only 62% reported finding it easy to consume this supplement. In addition, only 54% reported the 'shots' to be palatable. In another study, Babateen et al. [59] examined the feasibility of consuming low (1 beetroot juice 'shot' every second day), medium (1 beetroot juice 'shot' every day), and high (2 beetroot juice 'shots' every day) doses of concentrated beetroot juice in overweight/obese older adults in a 13-week intervention trial. Self-reported compliance was >90%. However, around 20% of the participants in that study withdrew prior to the end of the trial, with most of the dropouts coming from the moderate-or high-dose groups. Although some of the participants reported finding the taste of the beetroot juice 'pleasant', this was not a universal view, with others noting that the taste was 'bad' and that they had to 'hold [their] nose to gulp it down'. It is noteworthy that three of the participants who withdrew cited GI discomfort as their main reason for dropping out of the study, whilst three additional participants reported withdrawing because they disliked the smell/taste of beetroot juice.

In another study conducted in Tanzania, participants who consumed beetroot juice 'shots' for 60 days reported the taste to be 'good' or 'very good', and compliance with the intervention was >90% [60]. Nevertheless, the findings from both Ormesher et al. [58] and Babateen et al. [59] suggest that, at least for some individuals, more palatable alternatives to beetroot juice may be required to help some individuals elevate their nitrate intake. Other nitrate-containing supplements, such as beetroot-containing pills/capsules or nitrate-rich gels, could represent a more acceptable strategy to increase nitrate intake in individuals who do not like the taste of beetroot juice. In addition, the variety of options available with nitrate-rich vegetables (see Table 1 for some examples) could help reduce the risk of taste fatigue/monotony and allow individuals to self-select vegetables that fit better with their own food/taste preferences. This could, theoretically, help improve longer-term compliance.

Very high compliance (97–98%) was reported in 2 studies lasting 4–5 weeks, which used nitrate-rich vegetables as a nitrate source [61,62]. Meanwhile, in a study by Jonvik et al. [63], in which participants consumed four different nitrate-containing beverages—a sodium nitrate solution, concentrated beetroot juice, a rocket salad beverage, and a spinach beverage—ingestion of all four beverages was generally well tolerated. However, GI complaints were reported by two participants following consumption of the rocket salad beverage and by one participant following consumption of the spinach beverage, which was believed to be related to the high volume of these supplements provided (which was greater than for the beetroot juice and sodium nitrate conditions). In another study from the same group, participants increased their intake of dietary nitrate through either concentrated beetroot juice or whole vegetables over a 1-week period [64]. Both interventions were well tolerated, with no adverse events reported. Participants reported 100% compliance with the beetroot juice intervention, and consumed an average of 241 g/d nitrate-rich vegetables, which was close to their daily target of 250 g/d, suggesting that it is feasible to increase nitrate intake through both vegetables and nitrate-rich beetroot juice [64]. However, as noted by the authors, it is important to acknowledge that food and supplements were provided to participants in this study, such that different results may occur in the 'real world', where participants must purchase their own products [64]. It should also be noted that there may be some selection bias in the participants involved in these studies (i.e., individuals who like beetroot juice or vegetables may be more inclined to take part), such that these nitrate sources may be appraised less favourably in other cohorts.

One final point to note is that those choosing to use beetroot juice as a nitrate source should be aware of the potential for a slight discolouration of the urine, termed 'beeturia'. Although harmless, beeturia could cause shock/surprise (and, potentially,

unnecessary visits to a clinician) in some individuals if they are not forewarned about this side effect. Regardless, overall, it is clear that, at least in a research setting, it is feasible to elevate habitual nitrate intake using both whole vegetables and beetroot juice 'shots' over at least several weeks, and individuals looking to boost their nitrate intake could adopt either strategy, based around their own personal preference.

## 6. Efficacy

Dietary nitrate in the form of unconcentrated and concentrated beetroot juice [16,18,28,53,65,66], alongside whole vegetables—including, but not restricted to, whole beetroot [67], spinach [68], lettuce [69], mixed green leafy vegetables [70], and traditional Japanese vegetables, such as ta cai, chin gin cai, and garland chrysanthemum [71] (Table 3)—has been demonstrated to improve health parameters (especially blood pressure) and/or enhance exercise performance. This suggests that both juiced and whole-vegetable sources of nitrate can be effective at eliciting potentially beneficial effects, if a sufficient nitrate dose is administered [26].

Interestingly, several studies have compared the health and/or exercise performance effects of nitrate-rich beetroot juice against nitrate salts, and have demonstrated superior BP-lowering effects [63], greater reductions in oxygen consumption during steady-state exercise [72], and more pronounced attenuation of muscle pain after damaging exercise [73,74] with beetroot juice. Here, it is argued that the presence of compounds such as polyphenols and antioxidants in the beetroot juice may elicit effects synergistically to the nitrate (e.g., by potentiating nitrite reduction into NO and/or reducing NO degradation by mitigating superoxide scavenging) [75,76]. Similarly, McDonagh et al. [77] demonstrated greater BP-lowering effects of concentrated beetroot juice versus unconcentrated beetroot juice, a beetroot flapjack, and a drink containing nitrate crystals, providing further evidence that the form in which nitrate is consumed could impact its effects in the body. In contrast, there are few head-to-head comparisons between beetroot juice 'shots' and other vegetable sources of nitrate, which make it difficult to conclusively state whether one is superior (in terms of efficacy) to the other.

Some relevant evidence comes from a study by Jonvik and colleagues [63], who contrasted the effects of acute ingestion of concentrated beetroot juice against a rocket salad beverage, spinach beverage, and sodium nitrate solution (all containing 800 mg nitrate). In this study, all of the vegetable sources of nitrate were more effective at reducing BP than the sodium nitrate supplement, with no appreciable difference between the BP-lowering effects of concentrated beetroot juice and the spinach or rocket beverages. Interestingly, there were some differences in the plasma nitrate and nitrite response between supplements, which might suggest small differences in the optimal timing for ingesting each of the different nitrate sources, and requires further investigation. In another study by the same group, van der Avoort et al. [64] explored the BP-lowering effects of concentrated beetroot juice versus a variety of whole nitrate-rich vegetables (e.g., spinach, lettuce, arugula, bok choy) over a 1-week period. Although the concentrated beetroot juice resulted in greater increases in plasma nitrate and nitrite concentrations, both interventions were similarly effective at reducing systolic and diastolic BP (both by ~5 mmHg) [64]. Although additional head-to-head comparisons are required, from the limited evidence available, it is clear that both concentrated beetroot juice and vegetables can be effective at increasing nitrate intake and, consequently, impacting health parameters. Assuming similar doses are administered, it appears likely that the effects of dietary nitrate via concentrated beetroot juice and other vegetable sources are broadly comparable [26].

**Table 3.** Key intervention studies exploring the impact of nitrate-rich vegetables (excepting beetroot juice) on health and/or exercise performance parameters.

| Participants | Experimental Conditions | Duration | Key Findings | Reference |
|---|---|---|---|---|
| 19 healthy women aged 20 ± 2 years | Intervention: Diet rich in nitrate-containing vegetables, including lettuce, rocket, celery, leeks, fennel, and mixed salad leaves (nitrate dose: ~339 mg/d) Control: Diet low in nitrate-containing vegetables (nitrate dose: ~8 mg/d) | 1 week | Diet rich in nitrate-containing vegetables increased plasma nitrate and nitrite concentrations and decreased systolic BP by ~4 mmHg. | Ashworth et al. [78] |
| 30 (20 M, 10 F) participants with prehypertension or untreated grade 1 hypertension aged 63 (range: 56–71) years | Intervention 1: Increased intake of high-nitrate vegetables (~200 g/d, nitrate dose: ~150 mg) Intervention 2: Increased intake of low-nitrate vegetables (~200 g/d, nitrate dose: ~22 mg) Control: No increase in vegetables | 4 weeks | Intake of high-nitrate vegetables increased plasma nitrate and nitrite concentrations, but did not significantly impact BP or arterial stiffness. | Blekkenhorst et al. [62] |
| 30 (6 M, 24 F) healthy participants aged 47 ± 14 years | Condition 1 (control): Apple flesh (low in flavonoids, low in nitrate) Condition 2: Apple flesh and skin (high in flavonoids, low in nitrate) Condition 3: Apple flesh plus spinach (low in flavonoids, high in nitrate) Condition 4: Apple flesh and skin plus spinach (high in flavonoids, high in nitrate) | Acute intervention | Compared to the low-flavonoid, low-nitrate condition (condition 1), all treatments elevated plasma nitrite concentration, increased flow-mediated dilation, and reduced pulse pressure. Systolic BP was reduced by ~3 mmHg only in the high-flavonoid, low-nitrate (condition 2) and low-flavonoid, high-nitrate (condition 3) conditions. | Bondonno et al. [68] |
| 38 (12 M, 26 F) participants with normal to high BP aged 61 ± 7 years | Intervention: Diet rich in nitrate-containing vegetables, including frozen spinach and fresh lettuce, spinach, rocket, and other leafy green vegetables (nitrate dose: ~400 mg/d) Control: Diet low in nitrate-containing vegetables (nitrate dose: <100 mg/d) | 1 week | The high-nitrate diet increased plasma and salivary nitrate and nitrite. However, ambulatory, home, and clinic BP alongside arterial stiffness were unchanged with both the high- and low-nitrate diets. | Bondonno et al. [70] |
| 36 (14 M, 22 F) healthy participants aged 67 ± 6 years, | Intervention: 150 g whole cooked beetroot (nitrate dose: ~145 mg/d), plus 1 medium-sized banana every second day Control: 1 medium-sized banana every second day | 8 weeks | Beetroot consumption reduced systolic BP by ~8 mmHg and increased diversity (Shannon index) of the gut microbiota. | Capper et al. [67] |

**Table 3.** *Cont.*

| Participants | Experimental Conditions | Duration | Key Findings | Reference |
|---|---|---|---|---|
| 12 (6 M, 6 F) younger adults aged 27 ± 4 years, and 12 (3 M, 9 F) older adults aged 64 ± 5 years | Condition 1: 100 g cooked beetroot (nitrate dose: ~272 mg) Condition 2: 200 g cooked beetroot (nitrate dose: ~544 mg) Condition 3: 300 g cooked beetroot (nitrate dose: 816 mg) Condition 4: Potassium nitrate solution (nitrate dose: 1000 mg) | Acute | Plasma nitrate and nitrite concentrations increased in a dose-dependent manner, with more pronounced responses in younger vs. older adults. All interventions reduced systolic and diastolic BP (both by ~2–5 mmHg) compared to baseline in younger adults, whereas only the high-dose potassium nitrate intervention (condition 4) reduced systolic and diastolic BP (both by ~5 mmHg) in the older adults. | Capper et al. [79] |
| 18 (11 M, 7 F) healthy participants aged 18 ± 1 year | Condition 1: Sodium nitrate solution (nitrate dose: 800 mg) Condition 2: Concentrated beetroot juice (nitrate dose: 800 mg) Condition 3: Spinach beverage (nitrate dose: 800 mg) Condition 4: Rocket salad beverage (nitrate dose: 800 mg) | Acute | All conditions increased plasma nitrate and nitrite concentrations. The vegetable nitrate sources (conditions 2–4) reduced systolic (~5–7 mmHg) and diastolic (~4–8 mmHg) BP by a similar amount. The sodium nitrate did not impact systolic BP and led to smaller reductions in diastolic BP (~2–4 mmHg). | Jonvik et al. [63] |
| 27 healthy participants aged 24.5 ± 11 years | Condition 1: Spinach soup (845 mg/d nitrate) Condition 2: Asparagus soup (0.6 mg/d nitrate) | 1 week | The high-nitrate spinach soup reduced the augmentation index, lowered central systolic (−3.39 ± 5.6 mmHg) and diastolic BP (−2.60 ± 5.8 mmHg), and also reduced brachial systolic BP (−3.48 ± 7.4 mmHg) compared with the low-nitrate asparagus soup. | Jovanovski et al. [80] |
| 26 (6 M, 20 F) participants aged 59 ± 8 years | Intervention: High-nitrate meal (nitrate dose: 220 mg from spinach) Control: Low-nitrate meal | Acute | Consumption of the high-nitrate meal elevated salivary nitrate and nitrite, increased the large artery elasticity index, reduced pulse pressure, and decreased systolic BP (up to ~7.5 mmHg). | Liu et al. [81] |

| Participants | Experimental Conditions | Duration | Key Findings | Reference |
|---|---|---|---|---|
| 8 (5 M, 3 F) normotensive adults aged 73 ± 5 years | Condition 1 (control): 3 days low-nitrate diet (nitrate dose: 43.4 mg/d) Condition 2: 3 days low-nitrate diet + acute ingestion of nitrate-rich beetroot juice supplement (nitrate dose: 527 mg) Condition 3: 3 days high-nitrate diet (nitrate dose: 155 mg/d) Condition 4: 3 days high-nitrate diet + acute ingestion of nitrate-rich beetroot juice supplement | 3 days | Plasma nitrate and nitrite were significantly increased with the low-nitrate diet + supplement (condition 2) and high-nitrate diet + supplement (condition 4), but were unchanged in the other conditions. Systolic and diastolic BP were lower, compared to pre-breakfast levels, in all conditions. However, there was no difference in response between groups. | Miller et al. [82] |
| 7 healthy men aged 25 ± 2 years | Intervention: High-nitrate diet (nitrate dose: ~512 mg/d) including spinach and collard greens Control: Low-nitrate diet (nitrate dose: ~181 mg/d) | 6 days | The high-nitrate diet significantly increased plasma nitrate and nitrite concentrations, reduced the oxygen cost of moderate intensity cycling, increased total muscle work during intermittent leg-extension exercise, and improved repeated sprint ability test performance. | Porcelli et al. [51] |
| 25 (10 M, 15 F) healthy volunteers aged 36 ± 10 years | Intervention: Diet rich in traditional Japanese vegetables, such as ta cai, chin gin cai, and garland chrysanthemum (nitrate dose: 18.8 mg/kg/d) Control: Diet low in traditional Japanese vegetables | 10 days | The high-nitrate diet increased plasma and salivary nitrate and nitrite concentrations and reduced diastolic BP (~4.5 mmHg). Participants reported finding it easier to consume the diet rich in Japanese vegetables compared with the control diet. | Sobko et al. [71] |
| 231 (109 M, 122 F) adults with elevated BP aged 63 ± 6 years | Intervention 1: Low-nitrate vegetables + potassium nitrate pills (nitrate dose: 300 mg/d) Intervention 2: High-nitrate vegetables (nitrate dose: ~300 mg/d) + placebo pills Control: Low-nitrate vegetables + placebo pills | 5 weeks | Plasma nitrate concentrations increased with intake of potassium nitrate pills (intervention 1) and high-nitrate vegetables (intervention 2), whereas plasma nitrite only increased with the potassium nitrate pills. Ambulatory BP was no different between conditions, whereas with the potassium nitrate pills, there was a significant decrease in clinic diastolic BP post intervention. | Sundqvist et al. [61] |

**Table 3.** *Cont.*

| Participants | Experimental Conditions | Duration | Key Findings | Reference |
|---|---|---|---|---|
| 30 (15 M, 15 F) recreationally active participants aged 24 ± 6 years | Condition 1: High intake of nitrate-rich vegetables (nitrate dose: ~400 mg), including salad, lettuce, arugula, and beetroot<br>Condition 2: Concentrated beetroot juice (nitrate dose: 400 mg) | 1 week | Plasma nitrate and nitrite concentrations were elevated above baseline in both conditions, with greater increases in the concentrated beetroot juice condition. However, similar changes in systolic and diastolic BP (~5 mmHg reductions in both) were observed with vegetables and concentrated beetroot juice. | van der Avoort et al. [64] |

### 7. Other Benefits of Consuming Whole Nitrate-Rich Vegetables

For individuals who may be considering increasing their intake of dietary nitrate, doing so via consumption of a variety of whole nitrate-rich vegetables could have health benefits in addition to those associated with the nitrate, such as via the provision of various vitamins, minerals, and phytochemicals alongside dietary fibre [75,83] (an in-depth outline of the chemical/nutritional properties of beetroot juice can be found in [84,85]). Indeed, in a recent study by Capper et al. [67], ingestion of whole beetroot for 8 weeks reduced systolic BP by ~8 mmHg, and also increased the diversity of the gut microbiota and stool short-chain fatty acid concentrations. The increased (by ~7 g/d) intake of fibre with whole beetroot is likely to have contributed towards these effects [67,86,87]. Similarly, a recent animal model investigation suggested that intake of spinach, which is rich in nitrate, fibre, and a range of phenolics, can mitigate the impact of high-fat overfeeding on the gut microbiome and improve circulating glucose and lipid profiles [88].

### 8. Potential Barriers to Consuming Whole Nitrate-Rich Vegetables

To date, no studies have specifically investigated the barriers to dietary nitrate intake in the general public. However, it seems likely that barriers to general consumption of vegetables, such as convenience, availability, cost, knowledge, and time [89], may also be evident for consumption of nitrate-rich vegetables. Although concentrated beetroot juice overcomes some of these barriers (e.g., convenience and time), consumption of nitrate-rich vegetables may still represent a viable, cheaper, more palatable, and, therefore, potentially effective public health strategy for increasing dietary nitrate intake for a range of individuals.

Lack of education and knowledge about vegetable sources of dietary nitrate could be one potential barrier limiting intake of nitrate-rich vegetables. Increasing awareness of which vegetables are rich in nitrate may facilitate elevated intake in a range of individuals [90]. For example, it has previously been shown that better nutritional knowledge leads to better diet quality in both athletes [91] and the general population [92]. However, education alone is likely not sufficient to induce long-term lifestyle behaviour change [93,94], as dietary and other lifestyle changes can be incredibly difficult to make and maintain, and this must be considered at an individual, environmental, social, and policy level [95]. Whilst consideration for these broad strategies for behaviour change in this context are beyond the scope of this review, we consider herein some specific strategies that may be useful in optimising the intake of nitrate-rich vegetables.

### 9. Strategies to Help Optimise Intake of Nitrate-Rich Vegetables

For individuals considering increasing their intake of nitrate via whole vegetables, below we discuss some potential strategies that may be helpful and warrant further exploration in research and practical settings. Note, however, that the present paper exclusively addresses the scientific aspects of what is feasible and may be useful from a technical perspective and on a global scale. Many countries and regions have detailed rules and regulations on food health claims and marketing of foods with functional constituents, such as nitrate, and all countries tend to use the WHO guidance defining the ADI of nitrate at 3.7 mg/kg bodyweight [4], which (depending upon body weight) is close to the 400 mg nitrate per day used as a reference value throughout the present paper. The authors are aware that this affects, and in most cases limits, whether and how food producers and retailers can and should inform their customers about the nitrate contents of products, not to mention actively advertise the potentially associated benefits briefly reviewed in the Introduction. Due to the substantial differences in the relevant legislation across the world, any changes or harmonisations will require a combination of global and local initiatives in collaboration between scientific and legislative communities, which are outside the scope of the present paper. Therefore, these aspects are not specified or discussed at all.

### 9.1. Self-Monitoring Tools

Self-monitoring tools could be a useful strategy to allow individuals to monitor and, if necessary, adjust their intake of nitrate via vegetables and other whole-food sources. This could take the form of an app or other computerised tool, which could be used by individuals to estimate their intake of nitrate—something which many individuals do already for other nutrients/energy intake [96]. Comprehensive databases have now been developed on food sources of nitrate, which are beginning to be incorporated into commercially available dietary analysis software [97], which make this a realistic possibility in the near future. The integration of these tools with supermarket websites or online recipe books could be helpful for those who are not currently achieving their target nitrate intakes. Another self-monitoring strategy could include measuring biomarkers, which provide a more objective measure of nitrate intake and subsequent 'activation' into nitrite in the body. For example, relatively inexpensive test strips are now available, which allow individuals to monitor their salivary nitrite, and could be used to help 'fine tune' dietary nitrate intake [32].

Around 65–70% of ingested nitrate is excreted in the urine within 24 h of nitrate intake [98]. As such, urinary nitrate can also be used as an objective marker of habitual nitrate intake [3,99]. Although commercially available test strips are available, which can be used to measure urinary nitrate and the nitrite concentration in a free-living setting, individuals should be aware that measuring the nitrate concentration of spot urine samples may be less informative than collecting 24 h measurements (something that is likely to be burdensome and unrealistic for many individuals outside of a research setting). Moreover, urinary nitrate/nitrite concentrations can be altered as a consequence of different health conditions, such as urinary tract infections and kidney injury [100], which could confound interpretation of these values.

### 9.2. Growth of High-Nitrate Vegetables

As noted previously, a recent study by Qadir et al. [49] has demonstrated the potential to manipulate growing conditions to produce lettuce with a controlled high- and low-nitrate content. These products have potential for use as an alternative nitrate source and placebo (to nitrate-rich and -depleted beetroot juice) in a research setting [69]. In addition, if the high-nitrate vegetables (50 g of which would provide >500 mg nitrate) are mass produced at a relatively low cost, and approved for commercial sale, then they could represent one useful strategy to help individuals elevate their nitrate intake in a range of settings. Similarly, if the same growing methods could be applied to other vegetables, then this would open the possibility to provide a range of different high-nitrate vegetable sources that individuals could consume based on their preferred taste profile.

Urban agriculture (i.e., the growth of foods, such as fruits and vegetables, in urban areas) has attracted recent attention as a potential strategy to increase fruit and vegetable intake, reduce transportation costs, reduce stress, and increase the connectivity between people and the food system [44,101]. In densely populated cities, vertical farming units are one tool being employed in urban agriculture initiatives that allow highly standardised growing conditions [102], and could also be used to help ensure regular access to high-nitrate vegetables in specific community settings (e.g., work places, schools, retirement homes, prisons, and other locations where the provision of fresh vegetables poses logistical or economic challenges and intake is typically low).

### 9.3. Incorporation of Nitrate-Rich Vegetables into Existing Meals

Some individuals may find it difficult or objectionable to make major changes to the types of meals they consume daily (for example, swapping a favourite burger, pasta dish, or casserole for a nitrate-rich salad). In such cases, the incorporation of nitrate-rich vegetables into an individual's favourite foods or recipes (so called dietary 'hacks') could be a useful strategy to help 'boost' their habitual nitrate intake without requiring substantial modifications to the types of food they are eating. Covert incorporation of vegetables into meals has

been shown to be an effective strategy to enhance vegetable intake and can also have the parallel benefit of lowering the energy density of the meal [103]. This could be as simple as adding nitrate-rich vegetables, such as spinach, to a favourite soup or casserole. For example, one study demonstrated beneficial haemodynamic effects of a soup enriched with nitrate via spinach versus a lower-nitrate equivalent [80]. Another approach, which could be commercially exploited, is the development of baked products, in which nitrate-rich vegetables have been incorporated. The proof of concept for this idea was demonstrated in a series of studies by Hobbs et al. [104,105], who showed that beetroot-enriched bread was effective at elevating nitrate intake and reducing BP, particularly in T carriers of the Glu298Asp polymorphism in the eNOS gene. Importantly, the incorporation of beetroot into the bread did not appear to negatively impact the acceptability or palatability of the product [104]. A final dietary 'hack' that could help to increase nitrate intake could be the selection of cooking methods that minimise nitrate degradation/loss from foods, such as frying instead of boiling, as was employed in the DePEC trial in Malaysia [106].

*9.4. Integration into Dietary Guidelines*

Awareness of nitrate-rich vegetables could be increased via the incorporation of specific recommendations into relevant public health policies. For example, the UK-based Eatwell Guide recommends consumption of five portions of fruit and vegetables per day [107]; however, no consideration for the specific type of vegetables is provided. Specifying the number of portions that should come from nitrate-rich vegetables could be a useful strategy to increase public awareness and, therefore, intake of dietary nitrate. Similar concepts are currently adopted within the Eatwell Guide around the intake of fish, for which individuals are recommended to consume two portions per week, with one of these portions coming from an oily variety. Such recommendations could be implemented into other national policy recommendations, such as the US, in which specific recommendations for 'nitrate rich vegetables' could be added as a subgroup under the 'vegetables' food group [108]. It is important to note that it is still likely too early to make such recommendations, given the lack of long-term safety data regarding dietary nitrate intake, but this may be a feasible option in the future.

An alternative strategy to increasing consumption of nitrate-rich vegetables may be via a 'vegetables on prescription' approach. With this strategy, healthcare providers identify individuals or groups of individuals with low intake of fruit and vegetables (e.g., males, young people, low-socioeconomic-status groups, etc.), and prescribe fruit and vegetable intake through discounts on fruit and vegetable purchases [109]. This prescription is often delivered alongside messaging regarding the importance of fruit and vegetable intake. These interventions have had some success previously, improving knowledge related to the importance of fruit and vegetable intake [110]. This concept could be adopted for nitrate-rich vegetables, but the feasibility and efficacy of such interventions warrants investigation before implementation.

## 10. Conclusions

In this review, we have discussed the potential to use whole nitrate-rich vegetables as a strategy to increase dietary nitrate intake. We suggest that increasing consumption of whole nitrate-rich vegetables could be a useful and effective strategy for individuals seeking to elevate their habitual nitrate intake, either alone or alongside the beetroot juice 'shots' most commonly investigated in research. Whole vegetables are more cost effective than beetroot juice 'shots' in most contexts, and the potential to choose from a variety of vegetables to elevate nitrate intake could help avoid taste fatigue. In addition, numerous studies have shown that, providing a sufficient dose of nitrate is administered, whole vegetables can be an effective strategy to augment circulating NO metabolites and concomitantly improve health/exercise performance parameters. Nevertheless, it is our view that whole-vegetable sources of nitrate may not be a perfect replacement for beetroot juice 'shots' or equivalent concentrated and standardised products in all conditions, such as pre- or during exercise,

where the size (both for portability and in terms of volume required) and assured provision of a set nitrate dose may be advantageous. In addition, although other vegetable-based alternatives to the high- and low-nitrate beetroot juice 'shots' are beginning to emerge, these products are not yet widely available, such that the beetroot juice 'shots' are likely to remain the preferred nitrate donor in a research setting for some time yet.

Future research needs: In developed countries, different strategies should be explored, including self-monitoring tools, which can be used to help individuals optimise their nitrate intake via whole vegetables. Meanwhile, in developing countries, while the use of locally produced vegetables is more advantageous than imported beetroot concentrate, the most urgent need is to obtain more and better data on nitrate contents in the available vegetables. Together with the results from recent research on the benefits of dietary nitrate, such data can help to support producers and consumers to understand that high-nitrate content in vegetables may be an innocuous and even potentially beneficial trait, particularly when the overall vegetable intake is much lower than recommended.

**Author Contributions:** Conceptualisation, O.M.S.; methodology, A.G., S.A., E.J.H., J.M., K.B., A.W., M.S. and O.M.S.; software, not applicable; validation, not applicable; formal analysis, not applicable; investigation, A.G., S.A., E.J.H., J.M., K.B., A.W., M.S. and O.M.S.; resources, A.G., S.A., E.J.H., J.M., K.B., A.W., M.S. and O.M.S.; data curation, A.G. and O.M.S.; writing—original draft preparation, A.G., J.M., K.B., A.W. and O.M.S.; writing—review and editing, A.G., S.A., E.J.H., J.M., K.B., A.W., M.S. and O.M.S.; visualisation, E.J.H. and O.M.S.; supervision, O.M.S., K.B. and A.W.; project administration, O.M.S.; funding acquisition, not applicable. All authors have read and agreed to the published version of the manuscript.

**Funding:** This research received no external funding.

**Institutional Review Board Statement:** Not applicable.

**Informed Consent Statement:** Not applicable.

**Data Availability Statement:** Not applicable.

**Conflicts of Interest:** The authors declare no conflict of interest.

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
