# Peer review of "Exploring the Advantages and Disadvantages of a Whole Foods Approach for Elevating Dietary Nitrate Intake: Have Researchers Concentrated Too Much on Beetroot Juice?"

_applsci, doi:10.3390/app13127319_

Round 1

Reviewer 1 Report

This is a well-done review examining the comparison of elevating dietary nitrate intake by changing dietary patterns of wholefood consumption and also by consumption of beetroot juice.  Overall, the review is well done.  I only have a few substantive comments:

1) one drawback to the intake of changing patterns of wholefood consumption is that, well, it is hard to sustain such changes.  Another drawback that needs to be acknowledged is that the nitrate content of foodstuffs can be highly variable so knowing how much nitrate is being consumed is guesswork 

2) please comment on the ability to monitor nitrate or NO production in the urine with commercially available strips

3) the review focuses on a comparison to beetroot juice, however, beetroot pills/capsules are also commercially available.  This form of supplementation may be more palatable than beetroot juice and more sustainable over time than changing dietary patterns of wholefood consumption

4) on page 2 under the heading of "Availability" in the second paragraph, there is a statement that the highest rates of hypertension are in Africa and a WHO website is referenced.  However this statement is misleading.  I went to the WHO website and in their latest online report the highest rates of hypertension are in east Mediterranean followed by Europe.  Africa is below Europe.  Hypertension rates are high all over the world and while the quoted study in Tanzania is impressive, beetroot supplementation could be useful worldwide not just in geographic regions with the highest hypertension rates - which is not Africa

5) another potential use of beetroot supplementation is as an adjunct to hypertension pharmacotherapy.  Beetroot supplementation is not a recommended dietary change - though dietary change by augmenting more fruits and vegetables - is for sure is increasing nitrate ingestion.  However, a good case can be made to supplementing hypertensives with beetroot as an adjunct to treatment.

Author Response

Overall comment

This is a well-done review examining the comparison of elevating dietary nitrate intake by changing dietary patterns of wholefood consumption and also by consumption of beetroot juice.  Overall, the review is well done.  I only have a few substantive comments:

Author response:

Thank you for taking the time to review our manuscript and for your helpful feedback below.  We have responded to each point individually, and have modified the text in the manuscript accordingly. 

Specific comments

Comment 1:

One drawback to the intake of changing patterns of wholefood consumption is that, well, it is hard to sustain such changes.  Another drawback that needs to be acknowledged is that the nitrate content of foodstuffs can be highly variable so knowing how much nitrate is being consumed is guesswork

Author response:

Thank you for these valuable comments – we agree with both of the points here.  We have added additional text in the manuscript (see Section 3: Convenience, lines 183-184, and section 8: Potential barriers to consuming whole nitrate-rich vegetables, line 397) to further emphasise these points. 

Comment 2:

Please comment on the ability to monitor nitrate or NO production in the urine with commercially available strips. 

Author response:

Thank you for this comment.  We have added additional text to section 9.1, lines 436-445 to comment on the use of urinary test strips for monitoring NO bioavailability. 

Comment 3:

The review focuses on a comparison to beetroot juice, however, beetroot pills/capsules are also commercially available.  This form of supplementation may be more palatable than beetroot juice and more sustainable over time than changing dietary patterns of wholefood consumption

Author response:

Thank you for this comment – we agree, and we have now added additional text on this point in Section 5, lines 284-286.

Comment 4:

On page 2 under the heading of "Availability" in the second paragraph, there is a statement that the highest rates of hypertension are in Africa and a WHO website is referenced.  However this statement is misleading.  I went to the WHO website and in their latest online report the highest rates of hypertension are in east Mediterranean followed by Europe.  Africa is below Europe.  Hypertension rates are high all over the world and while the quoted study in Tanzania is impressive, beetroot supplementation could be useful worldwide not just in geographic regions with the highest hypertension rates - which is not Africa

Author response:

Thank you for highlighting this error.  We have adjusted the text to note that ‘High rates of hypertension’ rather than ‘The highest’ rates of hypertension are observed in Africa.  We agree that nitrate may have a role in combatting hypertension the world over, and our intention here is to simply emphasise one potentially underappreciated application of dietary nitrate in a setting which may not be familiar to some Western researchers.  

Comment 5:

Another potential use of beetroot supplementation is as an adjunct to hypertension pharmacotherapy.  Beetroot supplementation is not a recommended dietary change - though dietary change by augmenting more fruits and vegetables - is for sure is increasing nitrate ingestion.  However, a good case can be made to supplementing hypertensives with beetroot as an adjunct to treatment.

Author response:

Thank you for raising this point, which we agree with.  We have added additional text to the introduction noting that nitrate could be a useful adjunct to traditional antihypertensive treatments.  Please see lines 59-61.

Reviewer 2 Report

Comments 

The manuscript entitledExploring the advantages and disadvantages of a whole foods approach to elevating dietary nitrate intake: Have researchers concentrated too much on beetroot juice?” have explored the potential salutary effects of dietary nitrate in improving markers of cardiovascular and cognitive health, and enhancing exercise performance. The potential advantages and disadvantages of increasing consumption of various whole nitrate-rich vegetables to augment dietary nitrate intake has been discussed in detail. Numerous possible strategies which could be used to help individuals maximize their intake of nitrate via whole vegetables, and outline potential avenues for future research have also been focused.  The language needs to be modified and the following comments need to be addressed: 

Line 20, 21, 24, 28: Avoid spacing errors.

Line 20: Rewrite sentence

Line 42: Mention in detail which questions have been focussed in the present review.

Line 50-53: Mention in detail how consumption of nitrate can improve performance.

Line 89: How subheading 3.1, 3.2 can be after heading 2, Go through the manuscript and check numbering of each heading and sub heading 

Line 102: Avoid repetition; reference [4,32]. [4,32] is repeated, modify sentence   

Line 117-120: Rewrite sentence

Line 148-156: Add citation in the content

Line 157-167: Add citation in the content

Line 246: 249: Rewrite sentences

Line 267-278: Add citation

Line 336: Add more studies about benefits of consuming whole nitrate-rich vegetables

Line 473: Modify conclusion. Avoid writing citation in the conclusion, Conclusion should be as per the work reviewed by the author not what others have concluded and observed.

Line 568: Avoid writing old references, replace with the latest one

Line 570: Avoid writing old references, replace with the latest one

General comments: Modify language of the manuscript, modify typographical errors through the manuscript.

Keep uniform formatting while writing references throughout the manuscript 

Language of the manuscript needs to be modified

Author Response

General comments

The manuscript entitled “Exploring the advantages and disadvantages of a whole foods approach to elevating dietary nitrate intake: Have researchers concentrated too much on beetroot juice?” have explored the potential salutary effects of dietary nitrate in improving markers of cardiovascular and cognitive health, and enhancing exercise performance. The potential advantages and disadvantages of increasing consumption of various whole nitrate-rich vegetables to augment dietary nitrate intake has been discussed in detail. Numerous possible strategies which could be used to help individuals maximize their intake of nitrate via whole vegetables, and outline potential avenues for future research have also been focused.  The language needs to be modified and the following comments need to be addressed:

Author response:

Thank you for your comments and for taking to time to review our manuscript.  Please see below for a point-by-point breakdown of our responses and edits made to the manuscript. 

Specific comments

Comment 1:

Line 20, 21, 24, 28: Avoid spacing errors.

Author response:

Thank you for this comment. Although many advocate including 2 spaces after a full stop to improve readability, we appreciate that this approach is not universally accepted. We have therefore modified the manuscript to ensure consistent use of one space after each full stop. 

Comment 2:

Line 20: Rewrite sentence

Author response:

Thank you for your suggestion, we have now re-written this sentence for clarity.  Please see lines 20-23. 

Comment 3:

Line 42: Mention in detail which questions have been focussed in the present review.

Author response:

Thank you for this comment.  We have now clarified the specific questions of interest for this review at the end of the introduction.  Please see lines 88-99.

Comment 4:

Line 50-53: Mention in detail how consumption of nitrate can improve performance.

Author response:

Thank you for this comment, we have now added in details around how nitrate can improve exercise performance around lines 67-69.

Comment 5:

Line 89: How subheading 3.1, 3.2 can be after heading 2, Go through the manuscript and check numbering of each heading and sub heading

Author response:

Thank you for highlighting this error, this has now been rectified and we can confirm that all headings/sub-headings are correct. 

Comment 6:

Line 102: Avoid repetition; reference [4,32]. [4,32] is repeated, modify sentence  

Author response:

Thank you for highlighting this typographical error, which has now been amended.

Comment 7:

Line 117-120: Rewrite sentence

Author response:

Thank you for this comment, we have now rewritten this sentence for clarity (lines 150-156). 

Comment 8:

Line 148-156: Add citation in the content

Author response:

Thank you for your comment.  Appropriate citations have now been added to this section. 

Comment 9:

Line 157-167: Add citation in the content

Author response:

Thank you for this comment. We have now added appropriate citations to this section. 

Comment 10:

Line 246: 249: Rewrite sentences

Author response:

Thank you for this suggestion. We have re-written this sentence for clarity (lines 277-279).   

Comment 11:

Line 267-278: Add citation

Author response:

Thank you for this comment. We have now added appropriate citations to this section. 

Comment 12:

Line 336: Add more studies about benefits of consuming whole nitrate-rich vegetables

Author response:

Thank you for this comment.  We have now included additional supporting research in this section (lines 376-379). 

Comment 13:

Line 473: Modify conclusion. Avoid writing citation in the conclusion, Conclusion should be as per the work reviewed by the author not what others have concluded and observed.

Author response:

Thank you for this comment, we have now removed the citation from the conclusion. 

Comment 14:

Line 568: Avoid writing old references, replace with the latest one

Author response:

Thank you for this comment, we have now updated the reference. 

Comment 15:

Line 570: Avoid writing old references, replace with the latest one

Author response:

Thank you for this comment.  We have now added additional, more recent, references.   

Comment 16:

General comments: Modify language of the manuscript, modify typographical errors through the manuscript.

Author response:

Thank you for this comment, we have undertaken extensive proof reading of the manuscript and we have made a range of edits to remove typographical errors. 

Comment 17:

Keep uniform formatting while writing references throughout the manuscript

Author response:

Thank you for this comment, we have now corrected the inconsistent citations, which appeared to be an issue with our referencing software when transferring the text into the journal template. 

Reviewer 3 Report

Title: to elevate of for elevating

Abstract: add couple of sentences about the benefit of nitrate in human health

                  Check the language by an expert

Introduction add more studies about the side effects of dietary nitrate

Adjust section 2 format and convert the currency to USD

Line 89, is this sec 2.1? if not where sec 3

Tables 1 and 2 add column for references and add more plants enriched in nitrate

Column 3 in Tables 1 and 2, adjust the quantities to mg

Add the statistical analysis and sample size, replication, SD

Add table about the chemistry of beetroot juice (especially nitrogen compounds) and the availability of nitrate

Line 251 adjust citation

Table 3, move reference column as the last one

Make figure 1 to graphical abstract and move it from this location

Update references and check the style of references

Check the outputs of all references

Check the structure and grammar mistakes by an expert

Author Response

Thank you for your comments and for taking to time to review our manuscript.  Please see below for a point-by-point breakdown of our responses and edits made to the manuscript. 

Comment 1:

Title: to elevate of for elevating

Author response:

Thank you for this suggestion, we have now adjusted the title to read ‘…for elevating…’

Comment 2:

Abstract: add couple of sentences about the benefit of nitrate in human health

Author response:

Thank you for this comment.  We have now added additional information to the abstract on the health benefits of dietary nitrate (lines 18-20). 

Comment 3:

Check the language by an expert

Author response:

Thank you for this comment.  We have extensively checked the language of the manuscript and made modifications where necessary to improve clarity. 

Comment 4:

Introduction add more studies about the side effects of dietary nitrate

Author response:

Thank you for this comment. We have now added additional information on the potential side effects of dietary nitrate. 

Comment 5:

Adjust section 2 format and convert the currency to USD

Author response:

Thank you for the suggestion, we have now added in USD as well to maximise comprehensibility in an international audience.  We have still provided information in pounds sterling for a UK audience. 

Comment 6:

Line 89, is this sec 2.1? if not where sec 3

Author response:

Thank you for highlighting this error, we have now amended the section headings. 

Comment 7:

Tables 1 and 2 add column for references and add more plants enriched in nitrate

Author response:

Thank you for your comment.  We have now added additional nitrate containing plants. As the nitrate values are derived from one (Table 1) and two (Table 2) references only, we feel it is more appropriate to present the reference as a foot note to avoid repeating the same reference multiple times.  Therefore, we have not added a separate column for references. 

Comment 8:

Column 3 in Tables 1 and 2, adjust the quantities to mg

Author response:

Thank you for this comment.  We feel that it may be clearer to the reader if the amount of vegetable is presented in grams whilst the amount of nitrate is presented in mg, to avoid any confusion.  For clarity, we have made slight adjustments to the column titles. 

Comment 9:

Add the statistical analysis and sample size, replication, SD

Author response:

Thank you for this comment.  As this was a narrative review, no specific statistical analyses were conducted, therefore no edits have been made. 

Comment 10:

Add table about the chemistry of beetroot juice (especially nitrogen compounds) and the availability of nitrate

Author response:

Thank you for this comment. We agree that some readers may be interested to know more about the chemistry of beetroot juice.  However, we feel that it is beyond the scope of this article to comprehensively outline this here.  As such, we have referred the interested reader to in depth articles on this topic by leaders in the field.  Please see section 7, lines 371-372 for further details.  

Comment 11:

Line 251 adjust citation

Author response:

Thank you for highlighting this error, we have now amended the reference accordingly.

Comment 12:

Table 3, move reference column as the last one

Author response:

Thank you for this suggestion, we have now moved the reference to the last column.  

Comment 13:

Make figure 1 to graphical abstract and move it from this location

Author response:

Thank you for this suggestion, we have now made Figure 1 into the Graphical Abstract and moved it from the location as requested. 

Comment 14:

Update references and check the style of references

Author response:

Thank you for this comment, we have now updated the references and checked style to ensure consistency with journal guidelines. 

Comment 15:

Check the outputs of all references

Author response:

Thank you, we have checked and edited the references accordingly. 

Round 2

Reviewer 3 Report

Now can be accepted

The quality of language was improved